# In Situ Measurements of the Hydration Behavior of Compacted Milos (SD80) Bentonite by Wet-Cell X-ray Diffraction in an Opalinus Clay Pore Water and a Diluted Cap Rock Brine

Tobias Manzel [1,*], Carolin Podlech [1], Georg Grathoff [1], Stephan Kaufhold [2] and Laurence N. Warr [1]

1 Institute of Geography and Geology, University of Greifswald, 17489 Greifswald, Germany; carolin.podlech@uni-greifswald.de (C.P.); grathoff@uni-greifswald.de (G.G.); warr@uni-greifswald.de (L.N.W.)
2 Bundesanstalt für Geowissenschaften und Rohstoffe (BGR), 30655 Hannover, Germany; Stephan.kaufhold@bgr.de
* Correspondence: tobias.manzel@stud.uni-greifswald.de

**Abstract:** Compacted bentonite is currently being considered as a suitable backfill material for sealing underground repositories for radioactive waste as part of a multi-barrier concept. Although showing favorable properties for this purpose (swelling capability, low permeability, and high adsorption capacity), the best choice of material remains unclear. The goal of this study was to examine and compare the hydration behavior of a Milos (Greek) Ca-bentonite sample (SD80) in two types of simulated ground water: (i) Opalinus clay pore water, and (ii) a diluted saline cap rock brine using a confined volume, flow-through reaction cell adapted for in situ monitoring by X-ray diffraction. Based on wet-cell X-ray diffractometry (XRD) and calculations with the software CALCMIX of the smectite d(001) reflection, it was possible to quantify the abundance of water layers (WL) in the interlayer spaces and the amount of non-interlayer water uptake during hydration using the two types of solutions. This was done by varying WL distributions to fit the CALCMIX-simulated XRD model to the observed data. Hydrating SD80 bentonite with Opalinus clay pore water resulted in the formation of a dominant mixture of 3- and 4-WLs. The preservation of ca. 10% 1-WLs and the apparent disappearance of 2-WLs in this hydrated sample are attributed to small quantities of interlayer K (ca. 8% of exchangeable cations). The SD80 bentonite of equivalent packing density that was hydrated in diluted cap rock brine also contained ca. 15% 1-WLs, associated with a slightly higher concentration of interlayer K. However, this sample showed notable suppression of WL thickness with 2- and 3-WLs dominating in the steady-state condition. This effect is to be expected for the higher salt content of the brine but the observed generation of $CO_2$ gas in this experiment, derived from enhanced dissolution of calcite, may have contributed to the suppression of WL thickness. Based on a comparison with all published wet-cell bentonite hydration experiments, the ratio of packing density to the total layer charge of smectite is suggested as a useful proxy for predicting the relative amounts of interlayer and non-interlayer water incorporated during hydration. Such information is important for assessing the subsequent rates of chemical transport through the bentonite barrier.

**Keywords:** bentonite; waste repositories; smectite; swelling; hydration; water content; Milos; interlayers

## 1. Introduction

Bentonites are currently of interest as backfill material for the underground sealing of nuclear waste repositories. They are particularly suitable for engineering a multi-barrier system in hard crystalline rocks such as granite, where a low-permeability buffer material is required to fill the gap between the host rock and the radioactive waste containers [1–3]. The primary function of the buffer is to prevent or significantly slow down the rate of radionuclide transport in the case of leakage from the high-level nuclear waste containers. Due to the extremely low permeability of bentonites and their high cation exchange capac-

ity, these smectite-enriched materials are considered as suitable for retaining radioactive elements and thus preventing them from entering the host rock or biosphere [4,5].

The vital property of bentonite in forming a low permeability barrier is the ability of smectite to swell during hydration in aqueous solutions of varying electrolyte concentrations [6]. Hydrated bentonite can develop extremely low hydraulic conductivities in the order of $<10^{-11}$ m/s [7]. As bentonites are likely to be emplaced in the dry state either as compressed blocks or loser pellets (packing densities typically between 1.4 and 2.2 g/cm$^3$ [8]), it is important to study their hydration behavior once in contact with natural waters and to establish the mechanisms and rates of hydration during the early stages of saturation. Particularly relevant is to establish the mechanism by which water is stored in the bentonite and the way it affects the rate of subsequent chemical transport through the clay barrier.

There are three basic sites for non-crystalline water in hydrated bentonite [9]: (i) interlayer water adsorbed between two closely spaced negatively charged layers within smectite particles, (ii) adsorbed water on smectite particle surfaces and variably charged edge sites, and (iii) free pore water located in the spaces between grains. Bentonites dominated by interlayer water will be characterized by the lowest rates of chemical transport that reach the rates of diffusion [10,11] whereas bentonites with abundant free pore water will display higher rates of transport more characteristic of porous materials [12–14].

This study reports on the short-term hydration behavior of a Milos bentonite clay known as the SD80 sample, and evaluates its suitability when infiltrated by two types of solutions that simulate the natural groundwater of repository conditions in a mudrock formation and within a salt body. For this, the solutions used were an Opalinus clay pore water composition for the former, and a diluted cap rock brine for the latter. The hydration experiments were conducted as part of a long-term German research program known as the UMB (Umwandlungsmechanismen in Bentonitbarrieren) to aid the selection of the most suitable bentonite material in terms of their physical properties and mineral stability based on laboratory experimentation [12].

For the in situ experimental study of the hydration behavior of the SD80 bentonite, wet-cell diffractometry was used [9,15]. Compared to previously studied bentonites using this technique, the hydration patterns of the Milos bentonite shows some unusual features that may be related to some interlayer K as well as the generation of $CO_2$ gas from the dissolution of calcite. An assessment of related studies suggests the ratio of packing density to total layer charge provides a useful measure for predicting the hydration behavior of Ca- and Na-bentonites in advance of conducting a more detailed experimentation study.

## 2. Materials and Methods

For the experimental investigation, industrial bentonite from Milos, Greece (SD80) was used, for which the properties are well characterized (Table 1). Previous quantifications by X-ray diffraction (XRD) Rietveld analyses show the raw powder material contains 89% smectite and 11% accessory minerals such as feldspar and traces of quartz, calcite, pyrite, and baryte (Table 1). Assuming a pure dioctahedral nature, compositional analyses of the purified smectite fraction by energy dispersive X-ray analyses (EDX) produced a total interlayer charge of −0.36 e/phuc with −0.06 e/phuc distributed in the tetrahedral sheet and −0.30 e/phuc in the octahedral sheet [16]. The mineral formula calculation also indicates an interlayer content of 0.01 K, 0.03 Na, 0.09 Ca, and 0.06 Mg per half-unit cell (phuc), confirming Ca as the dominant exchangeable cation. The CEC of the bulk powder measured using the Cu-trien method [17,18] is 87 cmol/kg.

**Table 1.** Mineral assemblages and properties of SD80 bentonite compared to previously studied bentonite clays investigated by wet-cell X-ray diffractometry. Ant: anatase, Brt: baryte, Cal: calcite, Fsp: feldspar, Sme: smectite, Py: pyrite, Qz: quartz, n.d.: not determined, $\xi$: total layer charge, CEC: cation exchange capacity. IMA-CNMNC approved mineral symbols, after Warr [19].

| Sample | Sme | Fsp | Mca | Qz | Ant | Cal | Py | Brt | $\xi$ | CEC |
|---|---|---|---|---|---|---|---|---|---|---|
| | wt. % | wt. % | wt. % | wt. % | wt. % | wt. % | wt. % | wt. % | e/phuc | cmol/kg |
| SD80 (Ca-bentonite) [18] | 89 | 7 | n.d. | <1 | <1 | <1 | <1 | <1 | −0.36 | 87 |
| MX-80 (Na-bentonite) [20] | 76 | 5 | 3–4 | 5–6 | n.d. | <2 | <1 | n.d. | −0.28 | 70 |
| Ibeco-seal-80 (Na-bentonite) [21] | >80 | <3 | <3 | n.d. | n.d. | 8-12 | n.d. | n.d. | −0.33 | 82 |
| Tixoton-TE (Ca-bentonite) [21] | >80 | 5–6 | <2 | 8–9 | n.d. | <1 | n.d. | n.d. | −0.29 | 71 |

To evaluate the hydration behavior of the bentonite, two synthetic solutions were used as infiltrating fluids to simulate: (i) an Opalinus clay pore water (OPA), and (ii) a diluted cap rock brine (CAP) (Table 2). The Opalinus clay pore solution had a total salt concentration of 0.27 mol/L and a Na:Ca ratio of 8.1, whereas the diluted saline cap rock brine had a total salt concentration of 2.57 mol/L and a Na:Ca ratio of 78. The salinity of the two solutions was 19 g/L and 155 g/L, respectively, with starting pH values of 7.8 and 7.3 [12].

**Table 2.** Chemical composition of the Opalinus clay pore water (OPA) and the diluted cap rock brine (CAP) [12]. TDS = total dissolved solids.

| Dissolved Solids | Opalinus Clay Pore Water (OPA) [g/L] | Diluted Cap Rock Brine (CAP) [g/L] |
|---|---|---|
| NaCl | 12.39 | 145.87 |
| $CaCl_2$ | 2.89 | 3.55 |
| $Na_2SO_4$ | 1.99 | 5.40 |
| KCl | 1.27 | 0.37 |
| $MgCl_2$ | 1.62 | |
| $SrCl_2$ | 0.08 | |
| $NaHCO_3$ | 0.04 | |
| TDS | 20.23 | 155.19 |

Wet cells were used as confined volume reaction chambers to study the in situ hydration behavior of the SD80 bentonite powder [15]. A detailed description of the device and how it can be used to quantify the amount of interlayer and non-interlayer water uptake is given in previous publications [9,22]. An overview of the experimental set up is given in Figure 1. Before experimentation, the samples were equilibrated at laboratory conditions at 25 °C and ca. 50% relative humidity. In the SD80 experiments, the powder was introduced incrementally into the wet cell chamber and compacted using a metal cylinder of the same diameter. This way, relatively higher bulk densities of 1.48 and 1.50 g/cm$^3$ could be achieved. A thin X-ray transparent Kapton (polyimide) foil was used to cover the sample and reduce fluid evaporation (Figure 1a). This was fixed into place using a Teflon O-ring. Between XRD measurements, the wet cell was sealed with a Teflon lid that was held by a metal plate and screws. This ensured that a constant volume was maintained during hydration. For solution flow, two Teflon bottles were connected to each end of the wet cell using two-sided threads (Figure 1b). The upper bottle contained about 50 mL of the migrating solution and the bottom bottle was used to catch any percolating fluid. In the

case of the SD80 bentonite experiments, no water accumulated in the lower bottle. The amount of inflow solution was determined by regular weighing of the well-cell holder.

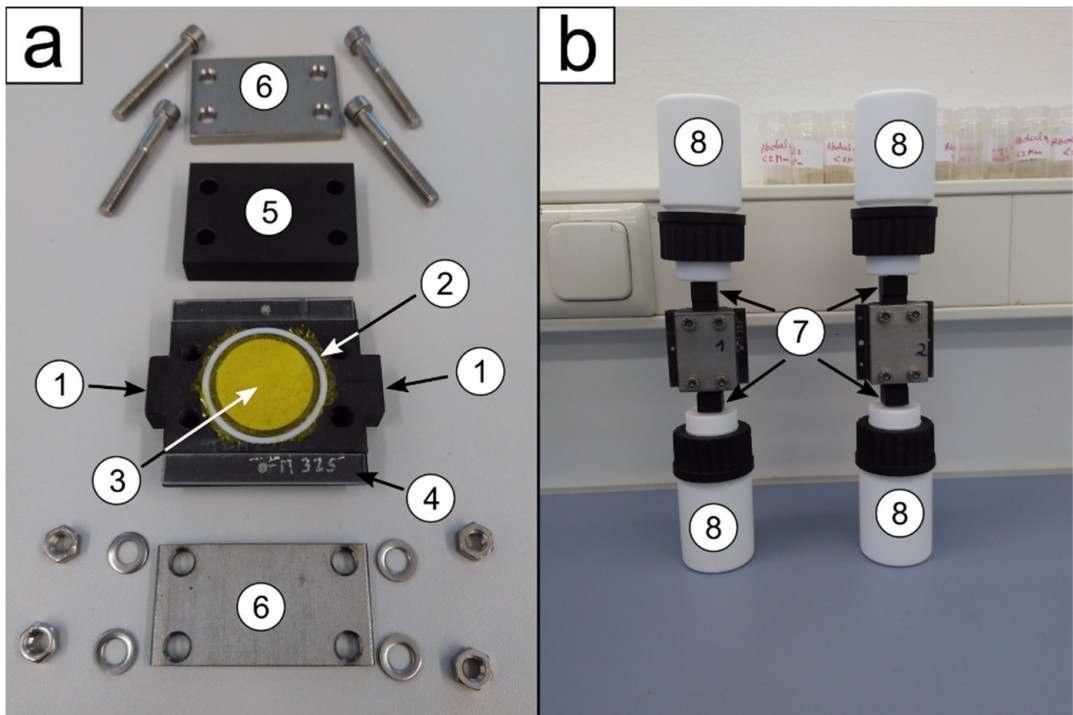

**Figure 1.** (**a**) Overview of the components of the wet cell. (**b**) Assembled wet cell assemblage. (1) Threaded inlets on each side of the cell. (2) Teflon O-ring seal. (3) Kapton foil (7.5 μm thick) used for sealing the sample chamber containing the compacted SD80 bentonite powder. (4) Modification bracket to mount the wet cell on the Bruker D8 Advance stage. (5) Teflon cover. (6) Metal plates and screws used to seal the cell with the Teflon cover to maintain a confined volume system. (7) Adaptors used to connect the wet cell and the Teflon bottles. (8) Teflon bottles to supply and capture the percolating solution.

For the XRD analysis, a Bruker D8 Advance diffractometer (D8 ADVANCE, Bruker, Billerica, MA, USA) with Fe-filtered CoK$\alpha$ radiation (40 kV, 30 mA) was used. The diffractometer was equipped with a 1D Lynx Eye Detector and a 0.5° divergence slit. The scans were collected from 3° to 50° 2θ with a scanning rate of 2° 2θ/min. The software CALCMIX (created by A. Plançon and V.A. Drits) [23], was applied to quantify the number of water layers (WLs) in the interlayer. For this, XRD patterns were matched using combinations of any three distinct WL configurations for smectite. The CALCMIX-calculated 001-reflection was adjusted to best fit the measured reflection by varying the percentage abundance of the specific WL configurations. It was assumed that the Reichweite $R$ is ordered and was set to $R = 1$ accordingly, and the particle thicknesses (the XRD-scattering domain sizes) had log-normal distributions with a mean around $N = 9$ as the average number of lattice layers. These assumptions represent a simplification of the variables used in previous studies [9,22], but comparisons of the sample XRD patterns used for the different studies produced consistent results within the errors of the method applied.

Based on the quantified percentages of the WLs and knowledge of the total amount of water inflow, it was possible to differentiate between interlayer water and non-interlayer water. Non-interlayer water is defined as the sum of adsorbed water and pore water not incorporated into interlayer sites. The interlayer water content was calculated as follows:

$$\sum_{n=1}^{4} n \cdot G_{wl} \cdot WL_n \tag{1}$$

where $n$ is equivalent to the number of water layers, $G_{wl} = 0.09$ cm³/g is the water content per layer [22,24–26], and $WL_n$ is the number of water layers derived from CALCMIX. The difference between the total intake monitored by weight change and the interlayer water content describes the amount of non-interlayer water content. As no surface area data was available for the SD80 bentonite, no attempt was made in this study to differentiate between the amounts of surface and pore water.

## 3. Results

While no visual changes were observed in the bentonite powders during the first 48 h, the monitoring of the weight change indicated some water uptake of around 0.05 mL/g (Figure 2).

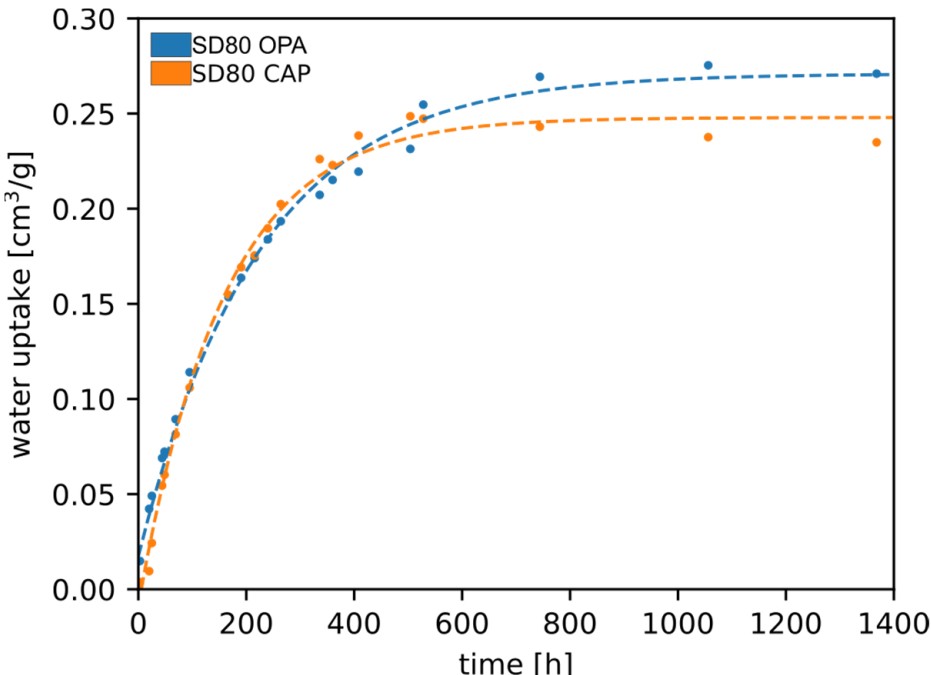

**Figure 2.** Comparison of the inflow of solution into the SD80 bentonite with Opalinus clay pore water (OPA), and diluted cap rock brine (CAP). Volumes were calculated from weight measurements after correction for salt content.

During the first 200 h of experimentation, both cells showed a similar rate of water intake with an average of $0.85 \times 10^{-3}$ mL/g·h. After 200 h, differences in the water uptake appeared, whereby SD80 with Opalinus clay pore water continued hydrating at a slow rate up to 1100 h before reaching its steady state. This was in contrast to the bentonite in diluted cap rock brine, which reached its maximum water uptake after 600 h. The slight decrease in water in the steady-state can be attributed to minor evaporation loss through the Kapton foil, which occurred at a rate that was faster than the inflow of water. At the end of the experimental run, when both SD80 samples of similar packing density reached a steady state of hydration, the sample reacted with Opalinus clay solution had taken in ca. 10% more water than the sample reacted with diluted cap rock brine (0.270 mL/g and 0.248 mL/g, respectively).

The first changes in sample appearance occurred after 167 h, when a radial darkening of samples occurred during water intake. This contrasts with the fast hydration front that crossed the samples observed by Warr and Berger [9] and indicates the wetting of the sample was slower and largely from below (Figure 3). That the Opalinus clay pore water sample showed clear darkening at the inlet indicated that a full wet state was not attained. In contrast, the diluted cap rock brine sample was fully darkened at the end of the experimental period, indicating a full wet state had been reached. Also, small gas bubbles

could be observed in the diluted cap rock sample trapped between the clay and Kapton foil. This feature was not observed in the SD80 cell infiltrated by Opalinus clay pore water.

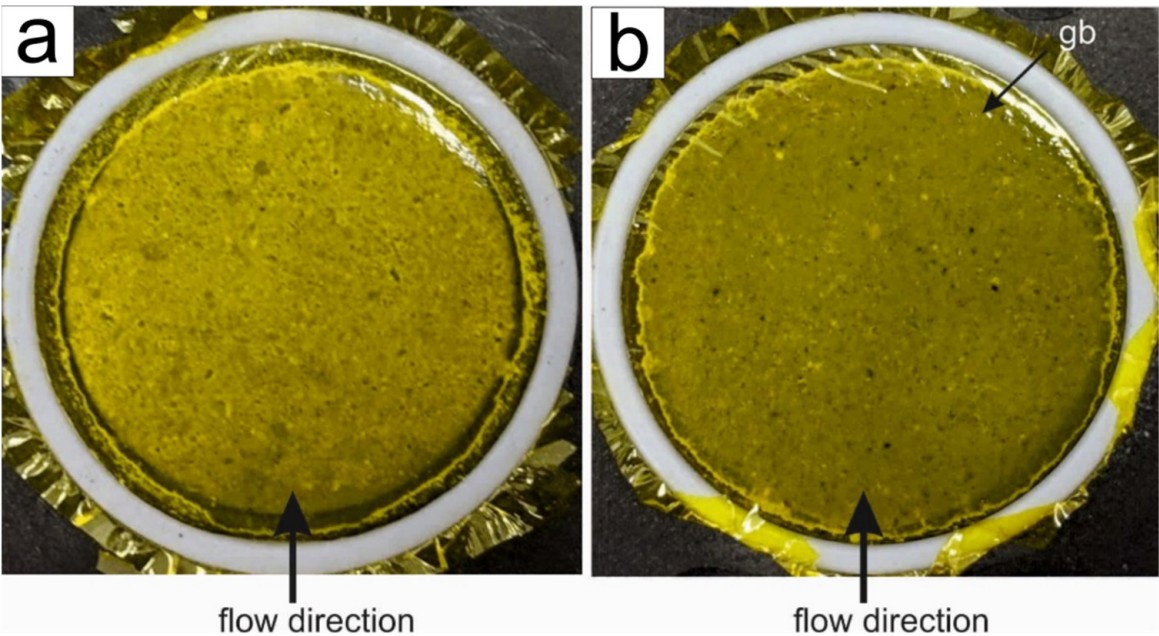

**Figure 3.** Photographs of the well cell experiments of hydrated SD80 bentonite close to the steady-state condition (**a**) infiltrated by Opalinus clay pore water and (**b**) infiltrated by diluted cap rock brine. gb = gas bubbles.

The measurement of the initial basal spacing of the dry SD80 powder determined by XRD before hydration was around 15 Å, as expected for a Ca-montmorillonite [27]. That indicates the dominance of the 2-WL structure in the interlayer under laboratory humidity conditions (ca. 50%). Modelling of the 001 basal planes using CALCMIX (Figure 4) indicated the 2-WL and 1-WL mixtures dominated the starting mixtures (Figure 5). The slight variation in abundance between the two samples probably reflected small differences in the air humidity at the time of measurement, whereby the wet cell that was infiltrated with diluted cap rock brine contained relatively less 2-WL and more 1-WL compared to the Opalinus clay solution-treated bentonite.

During the hydration of sample SD80 with the Opalinus clay solution, a rapid shift of the initial reflection towards higher basal spacing between 18 and 18.5 Å was observed after 69 h (Figure 4a). During the first 400 h of dehydration, this sample was characterized by the increase in 3-WLs at the expense of the less abundant 2-WL and 1-WL structures. After 450 h, when all 2-WLs had disappeared, some 3-WL structures grew to form 4-WL structures that varied in abundance between 18 and 25%. After 200 h, ca. 10% of 1-WLs remained intact during the remaining course of the experimentation and coexisted in a steady state with ca. 65% 3-WL and ca. 25% 2-WL.

A different pattern of interlayer hydration was observed in the diluted cap rock brine experiment (Figure 5b). In contrast to the first experiment, the basal spacing showed only a minor change towards large values with a shift of the initial reflection to a basal spacing of up to 15.2 Å (Figure 4b). CALCMIX calculations indicated the abundance of 2-WLs increased over the first 100 h and then stabilized at ca. 55%. This occurred at the expense of the 0 and 1 WLs, whereby the 0 WL structure only existed for the first hours of experimentation and the 1-WL structure decreased to a steady state of ca. 20%. The hydrated interlayer structure reached the steady-state condition after around 300 h with a mixture of ca. 55% 2-WL, ca. 25% 3-WL, and around 20% 1-WL. No WL structures >3 were detected.

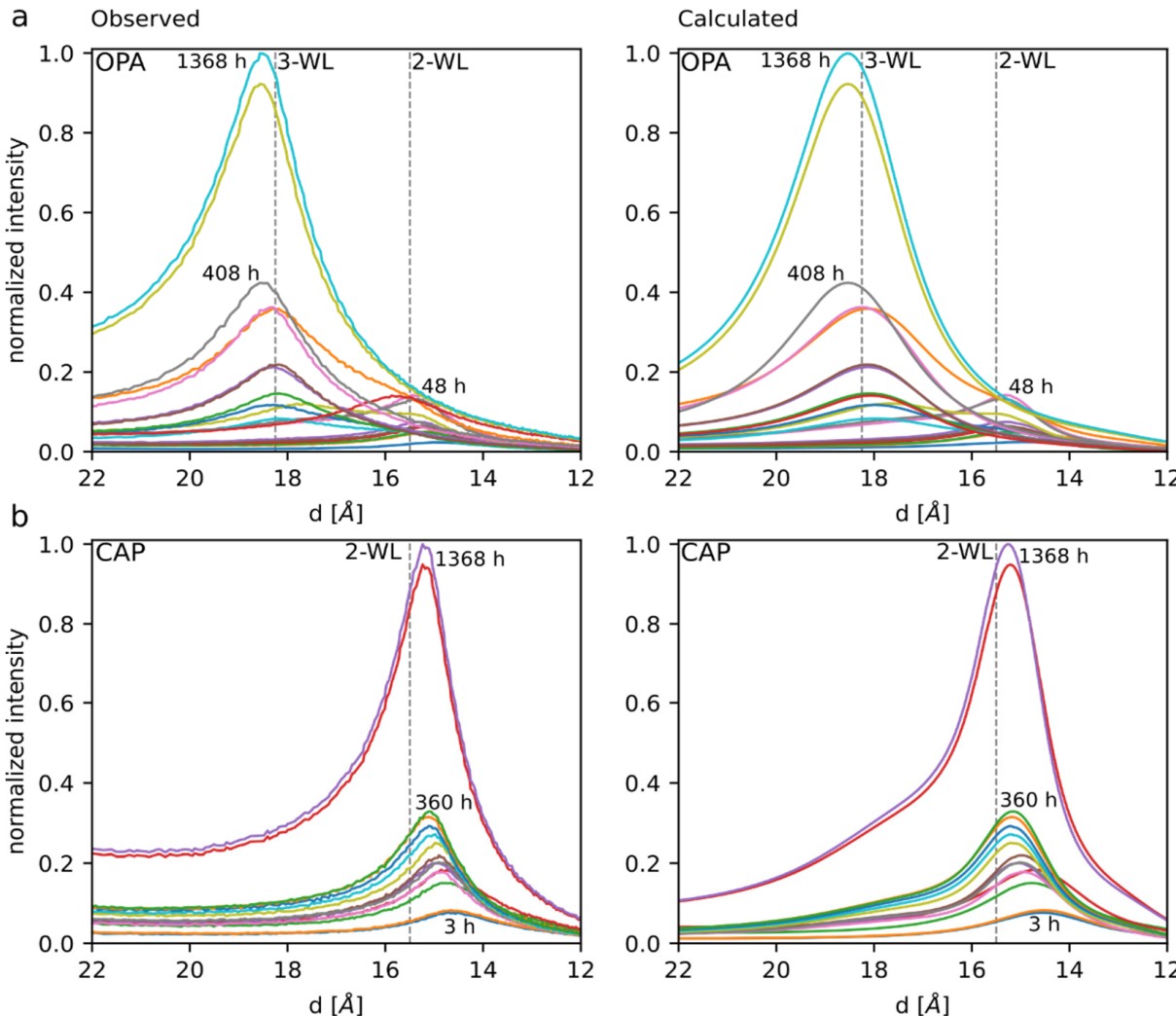

**Figure 4.** Comparison between observed and calculated XRD patterns for the varying hydration states of Milos bentonite. (**a**) SD80 infiltrated by Opalinus clay pore water, and (**b**) SD80 infiltrated by diluted cap rock brine.

Calculations of the amount of interlayer and non-interlayer water using Equation (1) showed major differences between the SD80 bentonite hydrated in the two types of solutions used. During the first 200 h when infiltrated by Opalinus clay pore water, the smectite incorporated a roughly equal amount of interlayer and non-interlayer water. After 200 h, more non-interlayer water instead of interlayer water entered the cell, resulting in a 57% and 43% mixture in a steady state. Overall, ca. 0.27 mL/g entered the SD80 bentonite by the end of the experiment.

The SD80 bentonite infiltrated with the diluted cap rock brine showed a similar rapid uptake of equal proportions of interlayer and non-interlayer water during the first 50 h of the experiment. However, after 50 h, the amount of interlayer water stopped at 0.05 mL/g and the uptake continued only as non-interlayer water. In the steady state, ca. 0.25 mL/g of total water content was comprised of ca. 80% non-interlayer water and just ca. 20% interlayer water. Compared with the Opalinus clay pore water hydrated sample, the diluted cap rock brine-infiltrated bentonite contained less than half the amount of interlayer water (0.05 mL/g compared to 0.12 mL/g).

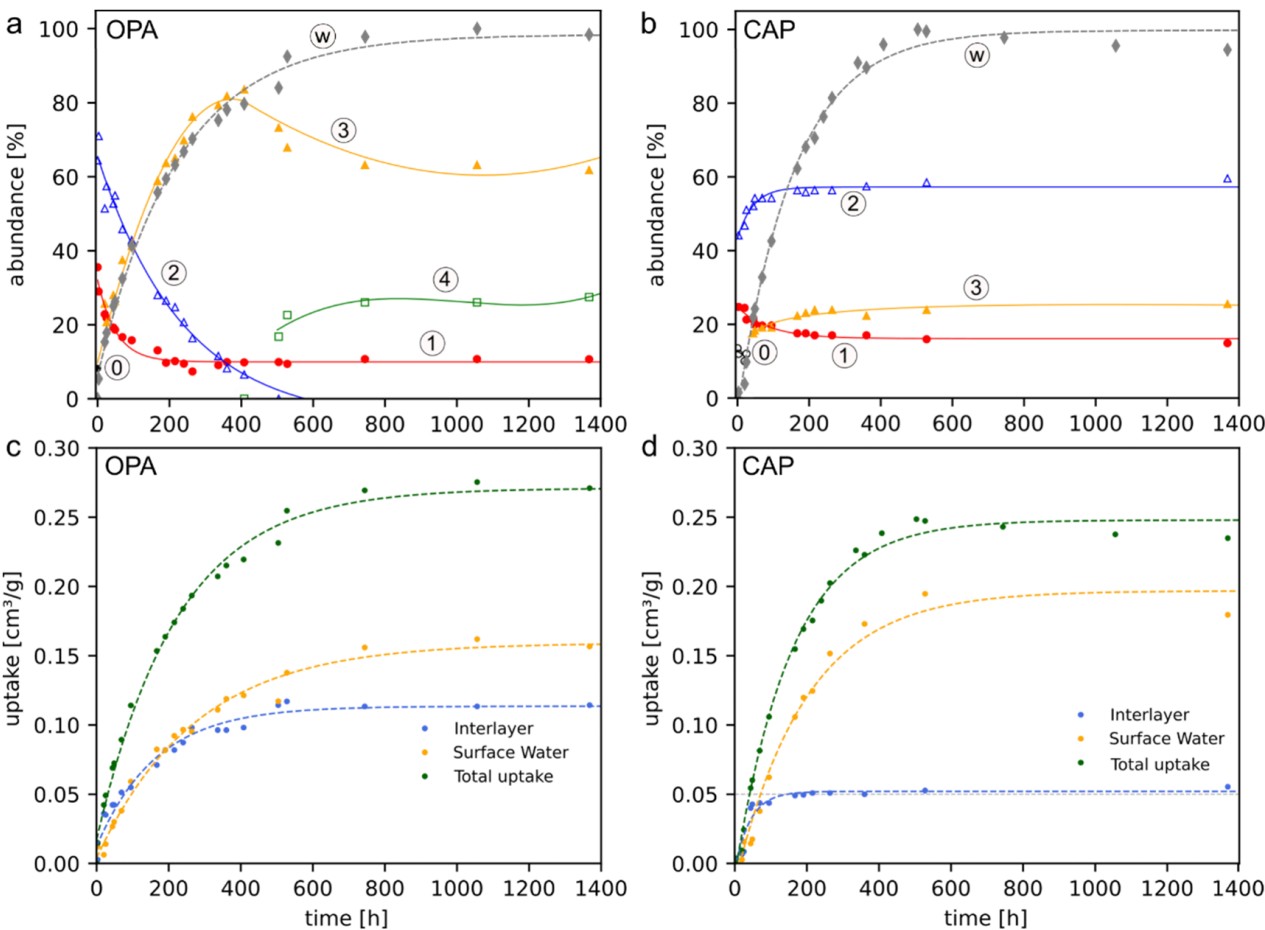

**Figure 5.** Relative abundance of water-layers developed (a,b) and the location of stored water (c,d) during the hydration of bentonite within the two wet cell experiments. The numbers of water layers (0, 1, 2, 3, 4) are shown in circles. (**a**) SD80 infiltrated by Opalinus clay pore water, (**b**) SD80 infiltrated by diluted cap rock brine, (**c**) partitioning of interlayer and non-interlayer (surface and pore) water in SD80 infiltrated by Opalinus clay pore water, (**d**) partitioning of interlayer and non-interlayer (surface and pore) water in SD80 hydrated by diluted cap rock brine. The total water uptake curve (w) is normalized to 100%.

## 4. Discussion

### 4.1. Hydration Mechanism of the Milos SD80 Bentonite

The overall hydration behavior of the Milos SD80 bentonite shows the expected pattern of interlayer expansion whereby the number of WLs increased until a steady state was reached. As the two hydration experiments were prepared with the same packing density and using the sample powdered material, the differences observed between the two wet cells can be solely attributed to the different chemistry of the infiltrating solutions used.

Considering the SD80 bentonite hydrated in Opalinus clay pore water (total dissolved solids (TDS) content = 19 g/L), the initial mixture of 2-, 1- and 0-WLs, expanded to 3- and 4-WLs. This is similar to the documented Ca-bentonite interlayer expansion in solutions of relatively high electrolyte concentration (13.9 g/L) where similar mixtures of 3-, 4-, and 2-WLs developed in a sample of low packing density [9]. However, the apparent disappearance of the 2-WL structure in the SD80 sample after 600 h while retaining ca. 10% 1-WLs is an unusual feature not seen in previously published experiments and requires explanation. As the CALCMIX program cannot model mixtures with more than three WL combinations, it may be partly an artifact, whereby the combination of 3-, 4-, and 1-WLs matched the patterns better than a mixture of 3-, 4-, and 2-WLs. Due to this limitation, some 2-WLs were likely retained in the steady state of hydration: albeit with a lower abundance

than the remaining 1-WL structures. The precise reason why 1-WL structures remained more abundant than 2-WLs in the Opalinus clay pore water once hydrated indicates that some of the interlayers in the Milos smectite were inhibited from swelling. One clear possibility could be the presence of small amounts of K (+0.03 e/phuc) in the interlayer sites, which are less prone to expansion and are likely to retain the 1-WL structure, even in the water-saturated state [28]. As ca. 8% of the layer charge (−0.36 e/phuc) was occupied by $K^+$, this corresponds well with the ca. 10% of 1-WLs remaining in the steady-state condition. The Opalinus clay pore solution contains 0.002 mol/L KCl, and given the high preference of $K^+$ adsorption to interlayer sites, some cation exchange is a possibility. Based on exchangeable cations as measured by the atomic adsorption spectroscopy (AAS) of SD80 smectite treated in long-term batch experiments at various temperatures, the amount of $K^+$ was lower than the unaltered smectite material with a ca. 20–40% decrease in concentration [16]. In contrast, the interlayer K content of untreated and treated bentonite measured by SEM-EDX analyses showed no significant difference between the two samples. These differences may reflect variations in the amounts of exchangeable and fixed K that occurred following experimental treatment.

The idea that the remaining 1-WL structures are related to some K cations in the smectite interlayer of the SD80 sample is also supported by the bentonite infiltrated by diluted cap rock brine. In this case, the higher abundance of 1-WLs remaining after hydration was over 15%, and this result corresponds with the higher amount of KCl in the solution (0.005 mol/L), which was 2.5 times more than in the Opalinus clay pore water sample. Composition measurements of the interlayer cations in the SD80 bentonite analyses following batch reactor experiments at 25 °C also showed less (20–40%) interlayer K following treatment with diluted cap rock brine [13,18]. SEM-EDX analyses of similar materials measured slightly higher (+0.02 e/phuc) amounts of interlayer K following treatment with the same brine [18]. The higher concentration of K remaining after hydration explains the higher abundance of remaining 1-WLs in this sample.

In contrast to the SD80 bentonite treated with Opalinus clay pore water, the same bentonite hydrated in diluted cap rock brine with a very high electrolyte concentration (TDS content = 155 g/L) showed a notably low degree of interlayer expansion, with a mixture of 2-, 3- and 1-WLs. As this is the most saline solution yet studied by wet-cell diffractometry, the strong suppression of thicker water layers and the high amounts of non-interlayer water in this experiment were likely to result from the high salinity of this brine. Such saline solutions significantly reduce the thickness of the double diffuse layer, and inhibit interlayer expansion by minimizing the difference in the concentration of ions in the interlayer and the surrounding pore water [6].

Another feature of interest was the occurrence of gas bubbles in the SD80 bentonite experiment hydrated by the diluted cap rock brine, which were trapped in solution between the upper surface of the bentonite clay and the Kapton foil. This can be attributed to the release of $CO_2$ gas associated with the dissolution of the calcite present in the bentonite sample. This gas was detected in batch reaction experiments conducted using the same material, and was responsible for generating significant gas pressures [12]. Batch reactions after 1 year produced some of the highest swelling pressures in diluted cap solution, reaching values of $2.21 \pm 0.01$ MPa [6].

The reason why gas bubbles were only observed in the more saline solution and not in the Opalinus clay pore water remains unclear. The intensity of the XRD reflections for calcite was notably higher than the sample containing the $CO_2$ bubbles, so it may have been due to heterogeneity of the material and small-scale differences in calcite abundance (both <1%). Furthermore, the pH of the diluted cap rock brine was slightly lower (pH 7.3) than the more alkane (pH 7.8) Opalinus clay pore water, but these small differences are unlikely to give rise to significantly different calcite dissolution rates and the pH was strongly buffered by the smectite clay in these experiments. One possible explanation could relate to the significantly higher Na:Ca ratio of the CAP solution (78) compared to the

Opalinus clay pore water (8.1) and its higher molar concentration of $Ca^{2+}$, which is known to increase the rate of calcite dissolution in Na–Ca–Mg–Cl brines (Tables 2 and 3) [29].

**Table 3.** List of samples with corresponding parameters from this study [1]; Warr and Berger [2], 2007; Perdrial and Warr [3], 2011; Berger, 2008 [4] [9,22,30]. BD = bulk density, DD = dry density, ξ = total layer charge, SSA = specific surface area, TWU = total water uptake, TDS = total dissolved solids, NIW = non-interlayer water. * = Na-smectite, # = Ca-smectite, + = estimation for Millipore water. n.d. = not determined.

| Sample | BD g/cm$^3$ | DD g/cm$^3$ | ξ phuc$^{-1}$ | SSA m$^2$/g | TWU mL/g | TDS g/L | Na:Ca Ratio | 1-WL [%] | 2-WL [%] | 3-WL [%] | >3-WL [%] | NIW mL/g |
|---|---|---|---|---|---|---|---|---|---|---|---|---|
| #SD80 [1] | 1.502 | 1.47 | −0.36 | n.d. | 0.27 | 19 | 8.1 | 10 | 0 | 63 | 27 | 0.16 |
| #SD80 [1] | 1.476 | 1.44 | −0.36 | n.d. | 0.25 | 155 | 78 | 15 | 60 | 25 | 0 | 0.20 |
| #TTE [2] | 0.94 | 0.76 | −0.29 | 103.01 | 0.61 | 0.041 | 0.3 | 0 | 13 | 42 | 45 | 0.46 |
| #TTE [2] | 0.94 | 0.78 | −0.29 | 103.01 | 0.69 | 13.6 | 26.4 | 0 | 7 | 63 | 30 | 0.53 |
| * IS80 [2] | 1.15 | 1.08 | −0.33 | 55.33 | 0.35 | 0.041 | 0.3 | 29 | 11 | 60 | 0 | 0.18 |
| * IS80 [2] | 1.14 | 1.04 | −0.33 | 55.33 | 0.45 | 13.6 | 26.4 | 0 | 55 | 40 | 5 | 0.23 |
| * +MX80 [3] | 1.43 | 1.41 | −0.28 | 30.03 | 0.40 | 0.0011 | - | 0 | 15 | 77 | 8 | 0.05 |
| * +MX80 [3] | 1.35 | 1.23 | −0.28 | 30.03 | 0.55 | 0.0116 | - | 0 | 0 | 61 | 39 | 0.18 |
| * +MX80 [3] | 1.60 | 1.31 | −0.28 | 30.03 | 0.34 | 0.0116 | - | 0 | 10 | 79 | 11 | 0.06 |
| * Swy-2 [4] | 1.37 | X1.28 | −0.32 | 27.64 | 0.44 | 0.0011 | - | 13 | 63 | 24 | 0 | 0.18 |
| * Swy-2 [4] | 1.36 | X1.27 | −0.32 | 27.64 | 0.45 | 58.44 | Na only | 22 | 64 | 14 | 0 | 0.19 |

Although the controlling parameters of $CO_2$ release remain uncertain, the generation of gas pressure in confined volume experiments is likely to influence the swelling behavior of the bentonite and may affect the thickness of the WLs developed. Compared to other bentonite experiments, the combination of the high swelling pressures produced by this clay and the limited interlayer expansion whereby the 2-WL was retained as the most abundant form may have been influenced by the additional effects of the $CO_2$ gas pressure. The interaction between gas generation and the swelling behavior of smectite in confined volume systems is a topic that requires further experimental study.

### 4.2. Predicting Bentonite Hydration in Confined Volume Systems Based on the Physical-Chemical Properties of the Bentonite

The mechanisms of smectite hydration are generally well understood. In addition to the effects of packing density and the volume of space available, the process is influenced by (i) the interlayer charge and its distribution, (ii) the type of interlayer cations present, and (iii) the chemistry and concentration of dissolved ions in the solution. After compiling all the available results of experiments conducted by X-ray wet-cell diffractometry (Table 3), it is of interest to discuss the overall patterns of hydration for variably compacted bentonites and the main controlling parameters. Based on this assessment, we consider whether or not a diagrammatic plot can be developed as a predictive tool for assessing bentonite hydration without actually conducting time-consuming experimental hydration tests.

Plotting the packing density versus the total water uptake reveals the expected trend of increasing packing density and decreasing water uptake (Figure 6a). The lowest amount of solution inflow was achieved in the two SD80 bentonite experiments of this study due to the higher dry densities used. This overall tendency appears to be largely independent of whether Na or Ca dominate the interlayer of the smectite. The type of interlayer cation is, however, a primary factor in determining the amount of interlayer versus non-interlayer water incorporated during hydration. Considering the relationships between the dominant interlayer cations of smectites (Figure 6b) shows clearly that relatively more non-interlayer water is incorporated into Ca-smectites than into Na-smectites, as established in previous studies [9,22]. These relationships appear to be largely independent of solution Na:Ca ratio, and the total content of dissolved salts indicates that any exchange reactions that occur during the hydration and wetting of the bentonite are far from incomplete. This is not unexpected given the very low solution-to-bentonite ratios involved in these experiments.

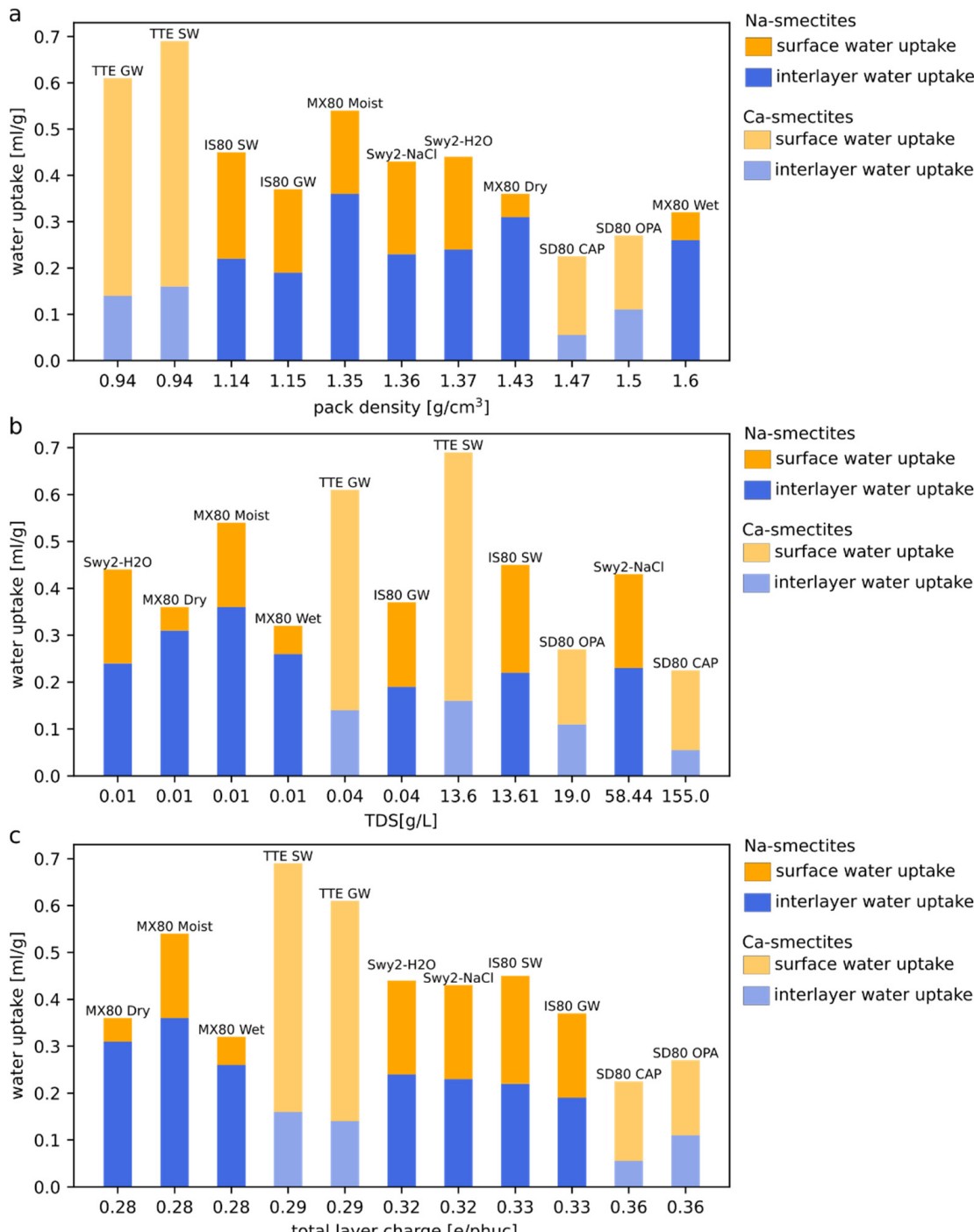

**Figure 6.** Summary of the amounts of interlayer and non-interlayer water in hydrated bentonite samples studied by wet-cell X-ray diffractometry [9,22,30]. (**a**) water uptake verses packing density, (**b**)water uptake verses total dissolved solids (TDS), (**c**) water uptake verse total layer charge.

The relationship between hydration and the total layer charge is less clear due to the small spread of charge values ranging between −0.28 to −0.36 e/phuc. The most water uptake occurs in Tixoton bentonite with a charge of −0.29 e/phuc and a decrease towards the higher layer charges of the SD80 bentonite. This pattern may, however, be partially explained by the differences in dry densities between these samples.

When considering the WL structures of all hydrated bentonite samples, some notable patterns can be recognized (Figure 7). For both Ca- and Na-smectites, higher dry packing densities lead to a lower number of WL structures due to the restriction of space, and, in the

case of Na-smectites, due to the build of higher swelling pressures caused by the additional osmotic gradients characteristic of Na interlayers (Figure 7a). As revealed by this study of the SD80 bentonite sample, which is dominated by Ca in the interlayer, the reduction in the thickness of the double diffuse layer in the brine and the additional generation of $CO_2$ are predicted to have a similar effect in repressing the thickness of the WLs that may develop. These effects have so far received little attention in previous swelling experiments.

In all Ca-smectites studied, the dominant thicknesses were 2 or 3 WLs, and the 2-WL was more common when using solutions with a high TDS content (Table 2; Figure 7b). In contrast, Na-smecties developed abundant 3 or >3 WLs, except for the purified Wyoming montmorillonite sample [30], where the very high smectite abundance of this clay did not develop thicker WL structures due to the higher swelling pressures generated when using a purified smectite sample. In this case, the Na-smectite of the Swy-2 samples in 1M NaCl (58.44 g/L) developed a very similar hydration structure to the Ca-smectite of the SD80 sample, with the dominance of 2-WLs and the preservation of some 1-WL structures. The precise role of the total layer charge on the WL structure remains uncertain from the interlayer hydration patterns (Figure 7c), although there is a tendency for the higher layer-charged smectites to retain fewer WLs compared to the lower-layer charges that generally favor thicker structures.

Despite the multitude of parameters that can affect WL structure and the partitioning between interlayer and non-interlayer water, the four most important physical–chemical features that can be used to describe the hydration behavior of bentonites can be plotted (Figure 8). By calculating the ratios of the interlayer water/total water uptake versus the packing density/total layer charge, general linear relationships are observed for both Ca- and Na-smectites. Bentonites dominated by interlayer hydration, high packing densities, and low layer charge plot more in the upper right part of the curve, whereas bentonites with high amounts of more mobile non-interlayer water, low packing densities, and higher layer charge fall in the lower left part of the curve. The plot also reflects how Na-bentonites are characterized by systematically higher interlayer/total water uptake ratios for any given state of packing density/total layer charge. As there are more data points for the Na-bentonite correlation ($R^2 = 0.91$, n = 7), this line is considered to be the more accurate of the two curves. With only four data points, the Ca-bentonite line ($R^2 = 0.68$) represents only a rough approximation and may well run quasi-parallel to the Na-bentonite correlation. Such a plot may be used as a useful predictive tool for assessing the mobility of water in hydrated bentonites whereby the upper-right parts of the curve represent low diffusion-controlled transport rates and the lower-left parts of the curve present faster rates of chemical transport due to the abundance of more loosely held surface and pore water. In terms of hydration behavior, the SD80 Milos bentonite is viewed as one of the more suitable backfill materials containing Ca-smectite, whereas the MX80 bentonite appears to be the most favorable Na-smectite variety considered here.

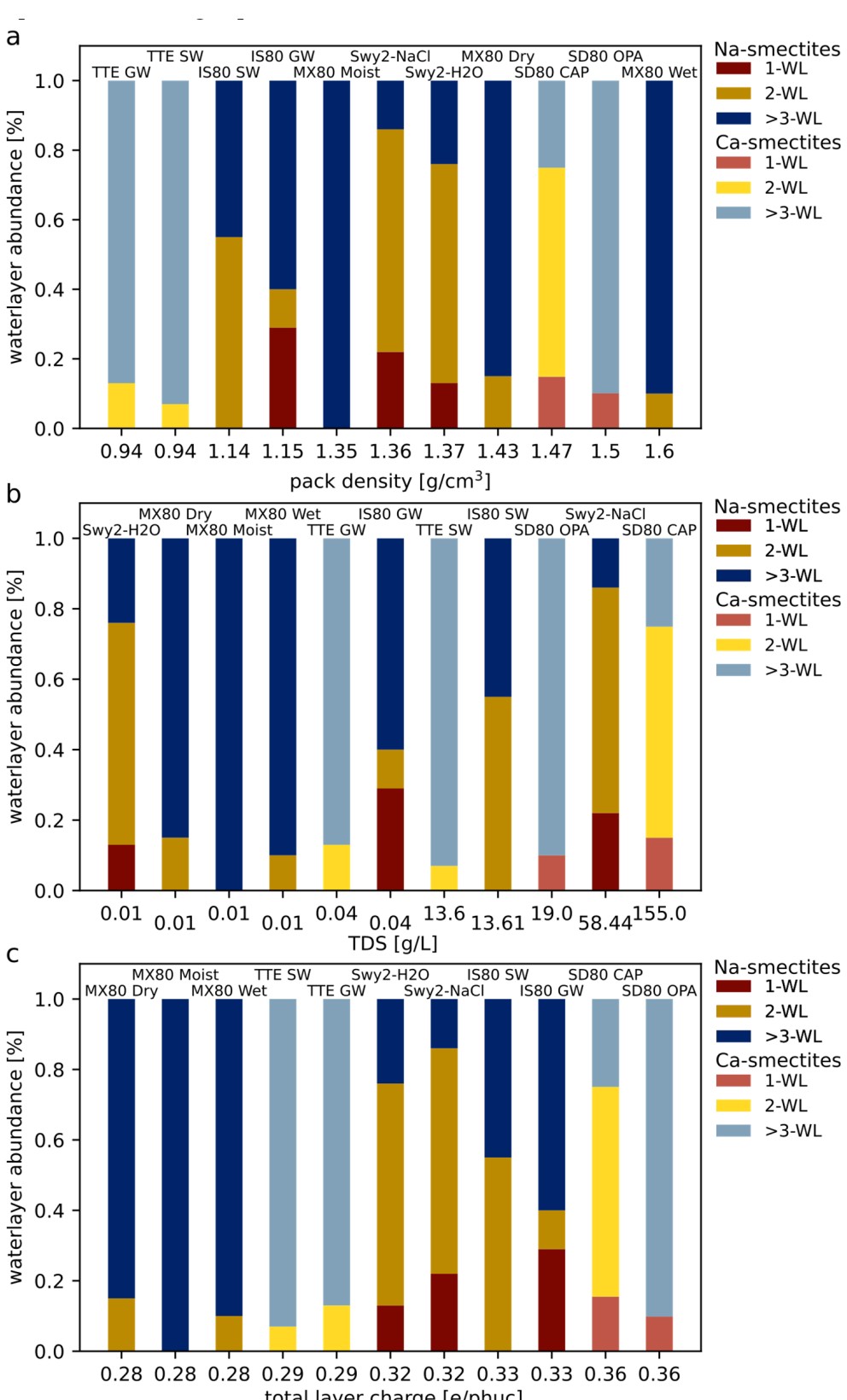

**Figure 7.** Summary of the number of water layers developed in hydrated bentonite samples studied by wet-cell X-ray diffractometry [9,22,30]. (**a**) water uptake verses packing density, (**b**) water uptake verses total dissolved solids (TDS), (**c**) water uptake verse total layer charge.

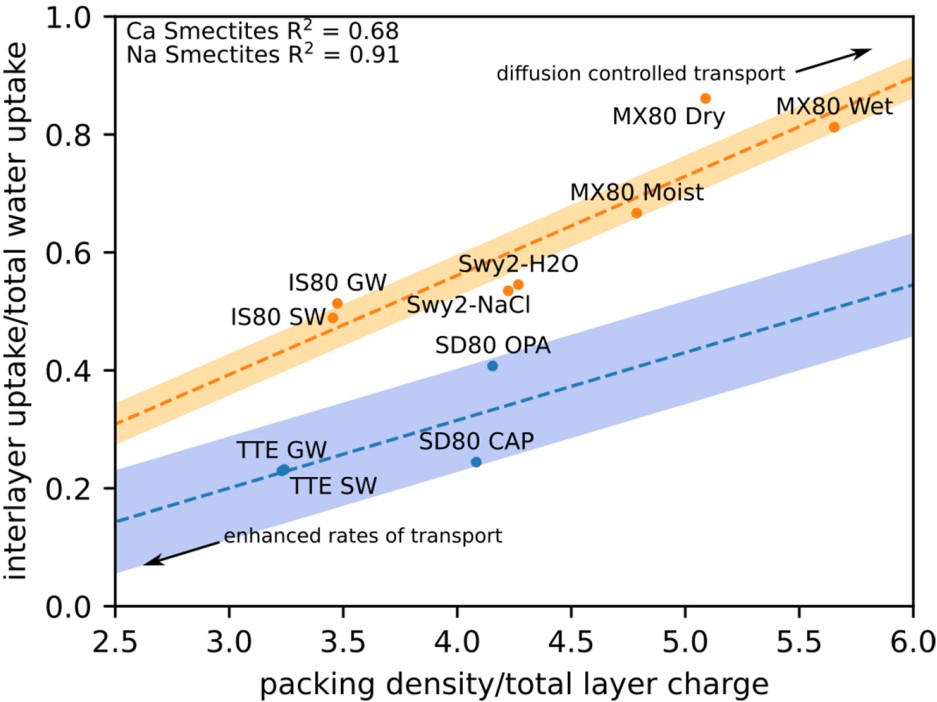

**Figure 8.** Plot showing the correlative relationships between the interlayer/total water uptake ratio versus the packing density/total layer charge ratios for Ca-(blue) and Na-(orange) bentonites (data source given in Table 3). The shaded areas correspond to the standard deviation of the data sets.

## 5. Conclusions

(1) The hydration of smectite studied by wet-cell X-ray diffractometry provides useful constraints for assessing the primary parameters of bentonite wetting that are controlled by the interacting parameters of the packing density, the total layer charge, and the type of interlayer cations.

(2) Hydration of the SD80 Milos bentonite shows some unusual features compared to published experiments. The retention of some 1-WLs in both the Opalinus clay pore water and the diluted cap rock brine is suggested to reflect K remaining in some interlayer sites. Also, the notably smaller thickness of WLs developed in the diluted cap rock brine probably resulted from the high total solid content of the solution, the generation of $CO_2$ bubbles, and the additional internal gas pressure within the bentonite clay, which is suggested to have additionally suppressed WL thicknesses.

(3) Whereas the chemistry and concentration of dissolved ions are also important, the overall patterns of bentonite hydration in many types of solutions indicate this factor to be more of a secondary effect during initial hydration and water saturation.

(4) For Ca- and Na-bentonite, determining the packing density/total layer charge ratio can be used as a predictive parameter for estimating the relative amount of interlayer-versus-non-interlayer water that will be stored in the system, which in turn will help assess the probable rates of chemical transport.

**Author Contributions:** Conceptualization, T.M. and L.N.W.; data curation, T.M. and C.P.; formal analysis, T.M. and C.P.; funding acquisition, G.G. and L.N.W.; investigation, T.M. and C.P.; methodology, T.M. and L.N.W.; project administration, G.G. and L.N.W.; resources, C.P. and S.K.; software, T.M.; supervision, G.G. and L.N.W.; visualization, T.M.; writing–original draft, T.M. and C.P.; writing–review and editing, T.M., C.P., G.G., S.K. and L.N.W. All authors have read and agreed to the published version of the manuscript.

**Funding:** This research is part of the joint project UMB, funded by the Federal Ministry of Economic Affairs and Energy (BMWi) under the grant number: 02 E 11344C. We also acknowledge support

for the Article Processing Charge from the DFG (German Research Foundation, 393148499) and the Open Access Publication Fund of the University of Greifswald.

**Data Availability Statement:** The data presented in this study are partially available on request from the corresponding author.

**Acknowledgments:** Special thanks to Artur Meleshyn, who provided helpful feedback.

**Conflicts of Interest:** The authors declare no conflict of interest.

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
