# Peer review of "In Situ Measurements of the Hydration Behavior of Compacted Milos (SD80) Bentonite by Wet-Cell X-ray Diffraction in an Opalinus Clay Pore Water and a Diluted Cap Rock Brine"

_minerals, doi:10.3390/min11101082_

Round 1

Reviewer 1 Report

The paper contained very interesting results and discussion. The whole paper is well wrote.  I recommend that the paper be published in the journal.

Author Response

Response to Reviewer 1 Comments

Manuscript minerals-1357714 entitled “In situ measurements of the hydration behavior of compacted Milos (SD80) bentonite by wet-cell X-ray diffraction in an Opalinus clay pore water and a diluted cap rock brine”

General Comments
Point 1: The paper contained very interesting results and discussion. The whole paper is well wrote.
I recommend that the paper be published in the journal.

Response 1: We thank the reviewer for the recommendation.

Reviewer 2 Report

In this study the authors have  examined and compared the hydration behavior of a Greek Ca-bentonite sample (SD80) in two types of simulated ground waters, 1) Opalinus clay pore water and 2) a diluted saline cap rock brine using a confined volume, flow-through reaction cell adapted for in situ monitoring by X-ray diffraction. Based on wet-cell X-ray diffractometry and software calculations of the smectite d(001) reflection, it was possible to quantify the abundance of water-layers in the interlayer spaces and the amount of non-interlayer water uptake during hydration using the two types of solution. The purpose was to address the use of compacted bentonite that is currently being considered as a suitable backfill material for sealing the underground repositories for radioactive waste as part of a multi-barrier concept. Although bentonites show favorable properties for this purpose, such as swelling capability, low permeability and high adsorption capacity, the best choice of material still remains uncertain. As they point out, There are three basic sites for non-crystalline water in hydrated bentonite: 1) interlayer water adsorbed between two closely spaces negatively charges layers within smectite particles, 2) adsorbed water on smectite particles surfaces and variable charged edge sites and 3) free pore water located in the spaces between grains. Bentonites dominated by interlayer water will be characterized by the lowest rates of chemical transport that reach the rates of diffusion whereas bentonite with abundant free pores water will display higher rates of transport more characteristic of porous materials. The authors undertook X-ray diffraction investigation of industrial bentonite from Milos, Greece following treatment with two separate infiltrating fluids. They found that the  hydration of smectite studied by wet-cell X-ray diffractometry provides useful constraints for assessing the primary parameters of bentonite wetting that are controlled by the interacting parameters of the packing density, the total layer charge and the type of interlayer cations. For Ca- and Na-bentonite, determining the packing density/total layer charge ratio can be used as a predictive parameter for estimating the relative amount of interlayer versus non-interlayer water that will be stored in the system, which in turn will help assess the probable rates of chemical transport.

            In my opinion the authors have done an excellent job of conducting very time-consuming experiments in order to examine the adsorption properties of bentonite. Their overall explanation and evaluation is well presented, and I suspect their work will be of extreme interest to the radioactive waste investigators. The references are excellent and the figures, for the most part, are very well displayed. However, I think it would be helpful when describing the design of the two Teflon bottles that were connected to each end of the wet-cell with two-sided threads to include a photo so the arrangement is clear to the average reader. Otherwise, I have just a few minor editorial suggestions to make:

Line 11 should read:….is currently being considered…

Line 58 should read:….two closely spaced negatively charged layers…

Line 89 should read:….also indicates an interlayer content…

Line 102 should read:...respectively, with starting pH values…

Line 314 should read:...chemistry and concentration of dissolved ions…

Line 339 should read:…solution to bentonite ratios involved…

Line 370 should read:…ratio for Ca-(blue) and Na-(orange) bentonites…

Reviewer 3 Report

Dear,

The goal of, In situ measurements of the hydration behavior of compacted Milos (SD80) bentonite by wet-cell X-ray diffraction in an Opalinus clay pore water and a diluted cap rock brine by authors,Tobias Manzel, Carolin Podlech, Georg Grathoff, Stephan Kaufhold  and Laurence N. Warr, was to examine  and compare the hydration behavior of a Milos (Greek) Ca-bentonite sample (SD80) in two types of  simulated ground waters i) Opalinus clay pore water and ii) a diluted saline cap rock brine using a  confined volume, flow-through reaction cell adapted for in situ monitoring by X-ray diffraction.

For Ca- and Na-bentonite, determining the packing density/total layer charge ratio can be used as a predictive parameter for estimating the relative amount of interlayer versus non-interlayer water that will be stored in the system, which in turn will help assess the probable rates of chemical transport.

Remarks:

  1. Line 18- You should be defined CALCMIX in abstract.
  2. Line 125- Explain the connection of the CALCMIX I ref 23.
  3. Reference No.19 is incomplete.
  4. Most of cited references are over ten years old.

Sincerely
